# PI-QT-Opt: Predictive Information Improves Multi-Task Robotic Reinforcement Learning at Scale

**Kuang-Huei Lee**[*]   **Ted Xiao**   **Adrian Li**   **Paul Wohlhart**   **Ian Fischer**   **Yao Lu**[*]

Google Research

[leekh, tedxiao, alhli, wohlhart, iansf, yaolug]@google.com

**Abstract:** The *predictive information*, the mutual information between the past and future, has been shown to be a useful representation learning auxiliary loss for training reinforcement learning agents, as the ability to model what will happen next is critical to success on many control tasks. While existing studies are largely restricted to training specialist agents on single-task settings in simulation, in this work, we study modeling the predictive information for robotic agents and its importance for general-purpose agents that are trained to master a large repertoire of diverse skills from large amounts of data. Specifically, we introduce Predictive Information QT-Opt (PI-QT-Opt), a QT-Opt agent augmented with an auxiliary loss that learns representations of the predictive information to solve up to 297 vision-based robot manipulation tasks in simulation and the real world with a single set of parameters. We demonstrate that modeling the predictive information significantly improves success rates on the training tasks and leads to better zero-shot transfer to unseen novel tasks. Finally, we evaluate PI-QT-Opt on real robots, achieving substantial and consistent improvement over QT-Opt in multiple experimental settings of varying environments, skills, and multi-task configurations.

**Keywords:** deep reinforcement learning, robot manipulation, multi-task learning

## 1 Introduction

Real robotic control systems are often partially observable and non-Markovian, and include high-dimensional observations, such as pixels. In such systems, we can learn representations by explicitly modeling the mutual information between consecutive states and actions – the *predictive information* [1] – to facilitate policy and value learning [2, 3]. When learning a predictive information representation between a state, action pair and its subsequent state, the learning task is equivalent to modeling environment dynamics [4]. In this work, we are interested in training multi-task generalist agents [5, 6, 7] that can master a wide range of robotics skills in both simulated and real environments by learning from a large amount of diverse experience. We hypothesize that modeling the predictive information will give latent representations that capture environment dynamics across multiple tasks, making it simpler and more efficient to learn a generalist policy. We also hypothesize that such a generalist agent may do a better job of transferring to real-world environments and novel tasks unseen during training. While existing studies of learning predictive information representations in RL [2, 3, 4, 8, 9] have largely been limited to learning single-task specialist agents in simulated environments such as DM-Control [10] and Atari [11], our hypotheses can be seen as extending the generalization results in Lee et al. [2], which showed both more sample-efficient learning and better fine-tuning on unseen tasks. We investigate both of these hypotheses in this work.

We combine a predictive information auxiliary loss with QT-Opt [12], a model-free off-policy reinforcement learning method that been shown to work well on vision-based continuous control problems. QT-Opt is able to leverage large-scale, multi-task datasets in simulation and the real world [12, 13, 14, 15]. We train a task-conditioned QT-Opt agent [12] with a predictive information auxiliary loss similar to Lee et al. [2], which we refer to as Predictive Information QT-Opt (PI-QT-Opt). We study various simulated and real robot environments using an Everyday Robots manipu-

---

[*]Equal contribution

6th Conference on Robot Learning (CoRL 2022), Auckland, New Zealand.

lator arm [16], including a large diverse set of real-world environments with up to 297 challenging vision-based manipulation tasks in a kitchen setting [7] (See Section 3.2 for task definitions).

Lee et al. [2] showed the benefits of predictive information regularization for accerlerating policy learning and quickly finetuning a policy trained on one task to solve a related task in the same environment. Our experiments show that predictive information regularization additionally gives substantial benefits in two challenging zero-shot settings: from simulation to real-world robots, and from one set of training objects to novel objects in both simulation and real environments. We demonstrate that PI-QT-Opt significantly outperforms QT-Opt in terms of success rate on training tasks in simulation. When evaluated on tasks that are unseen during training, modeling the predictive information increases the zero-shot success rate substantially. We verify these improvements on real robots via sim-to-real transfer, and observe that PI-QT-Opt significantly outperforms QT-Opt.

Our primary contributions are:

- We validate that modeling the predictive information is an effective auxiliary task for learning multi-task generalist robot control agents.

- We show that simple forms of task conditioning are sufficient to allow QT-Opt learn to solve large numbers of tasks simultaneously, avoiding some of the complexity of earlier multi-task approaches to QT-Opt [13].

- We verify the improvements through large-scale real-world experiments, including training a reward-based agent that performs well on 297 real robotic control tasks.

- We show that the predictive information helps zero-shot generalization to unseen tasks.

- We demonstrate that PI-QT-Opt can train only on simulated environments and transfer to real robots more effectively than our QT-Opt baseline.

## 2 Related Work

**Predictive Information Representations.** Previous studies [17, 2, 4, 3, 9, 18, 8, 19] have shown that predictive information [1] is an effective auxiliary or representation learning objective for RL agents or planning. This result can be connected to findings in neuroscience that suggested that the brain maximizes predictive information at an abstract level [20, 21]. Broadly, our work differs from those approaches by focusing on multi-task, vision-based robot learning in both simulated and real-world environments. This enables us to verify improvements and study broader generalization properties in realistic settings. Additionally, with the exception of [2], most such works do not explicitly learn a compressed representation of the predictive information. Finally, because the predictive information can model the underlying environment dynamics, which are stationary, its use as an auxiliary loss can also be seen as a representation regularizer for RL agents, possibly helping avoid overfitting to any specific value function during training [22].

**Robot Manipulation.** We focus on two main categories of related work in robot manipulation: multitask robot learning methods and real world data-driven robotics methods. One family of approaches for multitask robot learning focuses on supervised learning of multitask control policies to maximize few-shot performance via meta-learning [23, 24, 25] or direct zero-shot performance via behavioral cloning [26, 27, 28, 29]. However, methods based on imitation learning require expensive expert demonstrations and struggle to improve autonomously with on-policy learning. On the other hand, reinforcement learning (RL) based approaches are able to bootstrap without a large amount of expert data and are able to continuously improve from their own experience. While some RL methods combine subtasks [30, 31] or map tasks to individual policies [32, 33], we wish to learn a single shared policy. Recent work focuses on learning a single multi-task policy with both on-policy [34] and off-policy RL methods [35, 36, 37], and have shown promising results on a variety of robot manipulation tasks in simulation [34]. Some approaches to real-world robot learning for manipulation utilize RL with real robot data collection or simulation with domain randomization [15, 29, 12, 13, 38, 39, 40, 41, 42, 43], while other approaches focus on imitation learning from expert demonstrations [27, 44, 45, 46, 47, 48]. MT-Opt [13] is closely related to our approach; it also extends QT-Opt [12] to a real world multitask robot learning setting. While MT-Opt suggests that data routing and data sharing is very important, it is quite challenging to scale this approach to the hundreds of tasks that we consider. Thus, we take a different approach from MT-Opt, focusing on learning better representations for a single task-conditioned critic that does not use any data routing.

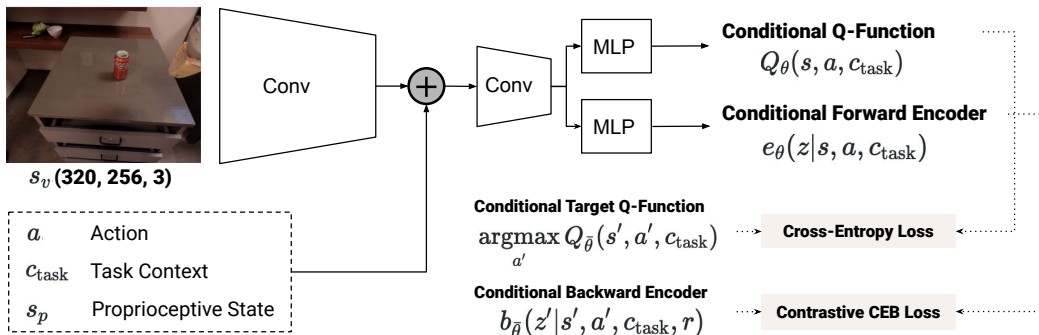

Figure 1: An overview of the PI-QT-Opt system for multi-task robotic reinforcement learning. The contrastive CEB loss is an auxiliary objective to Q-learning. $s = (s_v, s_p)$. See Section 3 for details.

## 3 Methods

We describe the details of Predictive Information QT-Opt (PI-QT-Opt) in Section 3.1, and task context conditioning for enabling multi-task robot RL at scale in Section 3.2. More implementation details are described in Appendix B. Figure 1 presents an overview of the our system.

### 3.1 Predictive Information QT-Opt

**Predictive Information.** The *Predictive Information* [1] is the mutual information between the past and the future, $I(past; future)$. It has been shown that the predictive information is an effective auxiliary loss for RL agents [2, 3]. From here on, we will denote the past by $X$ and the future by $Y$. Lee et al. [2] argues that a learned representation $Z$ of the predictive information should be compressed with respect to $X$, based on the observation in Bialek and Tishby [1] that $H(X)$, the entropy of the past, grows more quickly than $I(X; Y)$. Following [2], we use the Conditional Entropy Bottleneck (CEB) [49] to learn the representation $Z$, utilizing the same variational bound on CEB:

$$CEB \equiv \min_Z \beta I(X; Z|Y) - I(Y; Z) \tag{1}$$

$$\leq \min_Z \mathrm{E}_{x,y,z \sim p(x,y)e(z|x)} \beta \log \frac{e(z|x)}{b(z|y)} - \log \frac{b(z|y)}{\frac{1}{K}\sum_{k=1}^{K} b(z|y_k)} \tag{2}$$

where $(x, y)$ are sampled from the data distribution, $p(x, y)$, $e(z|x)$ is the learned *forward encoder* distribution, $b(z|y)$ is the learned variational *backward encoder* distribution, $\beta$ is a Lagrange multiplier that controls how strongly compressed the learned representation $Z$ is, with smaller values corresponding to less compression, and $K$ is the number of examples in a mini-batch during training. The second term of Equation (2) corresponds to the contrastive InfoNCE bound [3, 50] on mutual information $I(Y; Z)$. Following the CEB implementation of Lee et al. [51], we choose $e(z|x)$ and $b(z|y)$ to be parameterized von Mises-Fisher distributions. Details are described in Appendix B.1.

**QT-Opt.** QT-Opt is an offline actor-critic algorithm where only the critic is explicitly learned. It learns the Q-function (or critic) by minimizing Bellman errors:

$$\mathcal{E}(\theta) = \mathbb{E}_{(s,a,s') \sim p(s,a,s')} D\left[Q_\theta(s,a), Q_T(s,a,s')\right] \tag{3}$$

where $\theta$ is the set of model parameters, $s$ and $s'$ are state observations, $a$ is the action taken, $D$ is some divergence (QT-Opt uses cross-entropy), $Q_\theta$ is the learned state-action value function, and $Q_T = r(s,a) + \gamma V(s')$ gives a *target* value for the given transition $(s,a,s')$. $V(s') = \min_{i \in [1,2]} Q_{\bar{\theta}_i}(s', \pi_{\bar{\theta}_1}(s'))$ is a Double DQN state value function [52, 53, 54], which for QT-Opt is computed by using two lagged versions of the parameters $\theta$, $\bar{\theta}_1$ and $\bar{\theta}_2$, with different lagging methods. The QT-Opt policy, $\pi_{\bar{\theta}_1}(s) = \arg\max_a Q_{\bar{\theta}_1}(s,a)$, is optimized directly at each environment step using the cross-entropy method (CEM) [55]. In CEM, $N$ actions are sampled from a Guassian over the action space, the best $M < N$ actions as measured by $Q_{\bar{\theta}_1}$ are used to estimate the mean and variance of a new Guassian, from which another $N$ samples are drawn. This is repeated a

fixed number of steps, converging towards a narrow Gaussian over the part of the action space that the critic believes will perform best at the current state. See [12] for further details.

**PI-QT-Opt.** As shown in Figure 1, PI-QT-Opt combines a predictive information auxiliary similar to that introduced in Lee et al. [2] with the QT-Opt architecture. We define the past $(X)$ to be the current state and action, $(s, a)$, and the future $(Y)$ to be the next state, next optimal action, and reward, $(s', a', r)$. A state $s$ includes an RGB image observation and proprioceptive information. Image observations are processed by a simple conv net, the output of which is mixed with action, proprioceptive state, and the current task context (described in Section 3.2) using additive conditioning. A second simple conv net processes the combined state representation. All of the convolutional parameters are shared by both the forward encoder $e_\theta$ for modeling the predictive information (as in Equation (2)) and the Q-function $Q_\theta$ (as in Equation (3)), but the shared representation output from this is further processed by separate MLPs, to allow each loss to specialize its representation as needed, while still allowing the predictive information loss to influence the shared convolutional representation. Not shown in Figure 1 is that the target Q-function $Q_{\bar{\theta}_1}$ and backward encoder $b_{\bar{\theta}_1}$ for modeling the predictive information also share the same base lagged and non-trainable convolutional representation, but the backward encoder has its own trainable MLP, in order to learn any differences in dynamics when trying to predict the past from the future, rather than predicting the future from the past, as the forward encoder does. In addition, we concatenate the convolutional representation with observed reward $r(s, a)$ as the input to the backward encoder MLP head.

We find that adding a predictive information auxiliary loss is an easy way to give substantial performance improvements to our chosen RL algorithm, as in Lee et al. [2] which introduced Predictive Information Soft Actor-Critic (PI-SAC). However, we note that PI-SAC on its own was unable to solve our tasks, yielding close-to-zero success rates, and neither was SAC [56][2], which may indicate that the choice of base RL algorithm is still critical.

### 3.2 Multi-Task Context Conditioning

In order to learn one general-purpose agent for multiple tasks, we condition the Q-functions and the Predictive Information auxiliary on a *task context*, which describes the specific task that we wish the agent to perform, as illustrated in Figure 1. In our setting, a task involves a robot skill and a set of objects that the robot should interact with. We use two practical implementations of task context in different robot manipulation settings (Section 4.1). One is image-based, where a task is specified with the initial image, the initial object locations, and the skill type as depicted in Figure 2. It only considers locations and skill types and thus could enable good generalization across different and even novel objects. The other one is language-based, where tasks are specified with natural language, similar to Ahn et al. [7]. Details of these two implementations are described in Appendix B.2. A generalization of the language-based approach is considered in Appendix F.

An alternative approach to extending QT-Opt for multi-task learning is having one critic head per task with a shared base encoder, as used in MT-Opt [13]. However, MT-Opt found that a multi-headed architecture performed worse than a single-headed architecture; in addition, we note that a multi-headed strategy is practically prohibitive to scale to the order of hundreds of tasks.

## 4 Experimental Setup

To analyze how PI-QT-Opt compares with QT-Opt across different multi-task robotic learning scenarios, we explore a variety of challenging simulation and real vision-based robotic manipulation environments. While prior results on large-scale robotic grasping focused on a limited set of tasks [12, 13], we verify the robustness and scalability of PI-QT-Opt by studying many different environments across hundreds of different tasks in the real world.

### 4.1 Robot Manipulation Tasks

We study 6 different multi-task, vision-based robotic manipulation settings in 3 different environments in simulation. Four of the manipulation settings have the corresponding hardware setup permitting real-world evaluation.

---

[2]With our best effort, we were unable to get SAC and PI-SAC working on our tasks (See Appendix B.4)

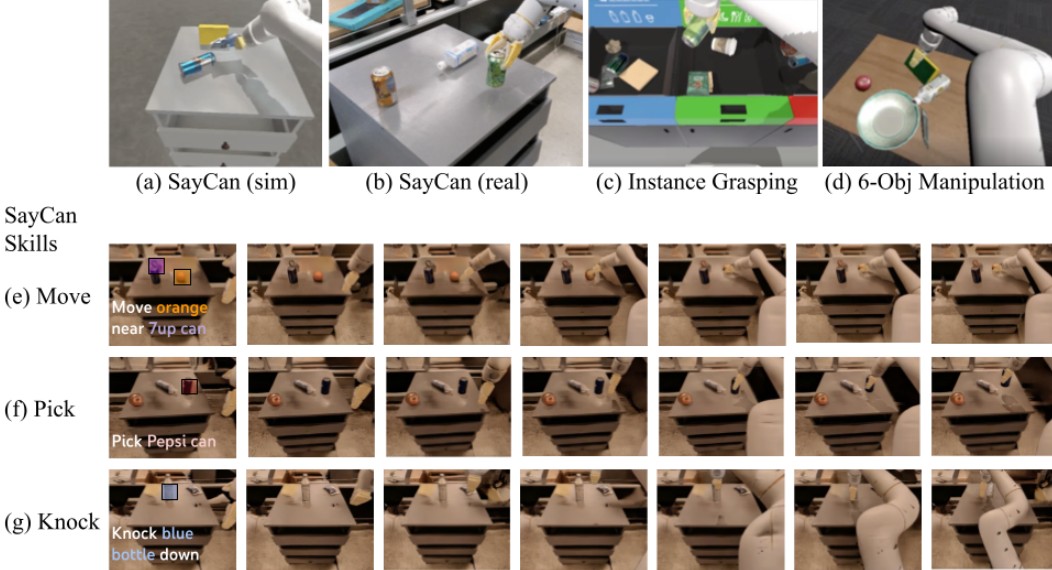

Figure 2: Example tasks in simulation and real-world environments using the Everyday Robots manipulator arm. See Section 4.1 for details. The first images of (e), (f) and (g) show target object locations used in the image-based task context conditioning method we designed. In real-world environments, the object locations are detected using a vision model (Section 5.2). An overlay image is created based on the object locations and the skill type, where we use different colors to indicate different skills. This overlay image is only created for the first image of an episode. Then, the first image and the overlay image are used as task context throughout the whole episode.

*Experiment 1:* **(Sim & Real) SayCan Move skill** (Figure 2(a),(b),(e)). We use the kitchen environment from SayCan [7] which contains 17 objects that spawn on a countertop. The simulation environment is shown in Figure 2(a) and the real countertop is shown in Figure 2(b). The Move skill contains 272 tasks of the form "move object A near object B". 246 tasks are used during training and the 26 remaining tasks are held out. Image-based task conditioning is used.

*Experiment 2:* **(Sim & Real) SayCan Pick skill** (Figure 2(a),(b),(f)). Same environment as Experiment 1. The Pick skill encompasses picking up each of the individual objects, for a total of 17 tasks. 12 tasks are used during training and the remaining 5 tasks are held out. Image-based task conditioning is used, as described in Section 3.2.

*Experiment 3:* **(Sim & Real) SayCan Knock skill** (Figure 2(a),(b),(g)). Same environment as Experiment 1. The Knock skill contains 8 tasks testing knocking over a can or bottle. 7 tasks are used during training and 1 task is held out. Image-based task conditioning is used.

*Experiment 4:* **(Sim & Real) SayCan 297 tasks, All skills** (Figure 2(a),(b),(e)-(g)). Same environment as Experiment 1. This task set includes all 3 SayCan skills (297 tasks). 265 tasks are used during training and the 32 remaining tasks are held out. Image-based task conditioning is used.

*Experiment 5:* **(Sim Only) Instance Grasping** (Figure 2(c)). A sampled subset of 37 different trash objects are placed randomly in bins. A vision model provides a target object for the robot to grasp. The episode is successful if the robot lifts the target object. Image-based task conditioning is used.

*Experiment 6:* **(Sim Only) 6-Object Manipulation** (Figure 2(d)). A fixed set of 6 objects are placed randomly on a table, similar to [27]. There are 30 separate tasks comprising the manipulation skills of picking, pushing, and pick and place. Language-based task conditioning is used.

In all of the experiments, we use an Everday Robots manipulator robot [16] with parallel-jaw grippers, an over-the-shoulder camera, and a 7-DoF arm. The robot has a proprioceptive observation space that includes the RGB camera image, the arm pose, and the gripper angle. For the action space, learned policies control the robot via relative position control of the end effector. In the simulation-only environments, we utilize blocking control, where the policy waits until the previous action completes before planning the next action. Motivated by faster and more reactive robot motions for real world evaluations, we utilize concurrent control [14] in all SayCan experiments, which

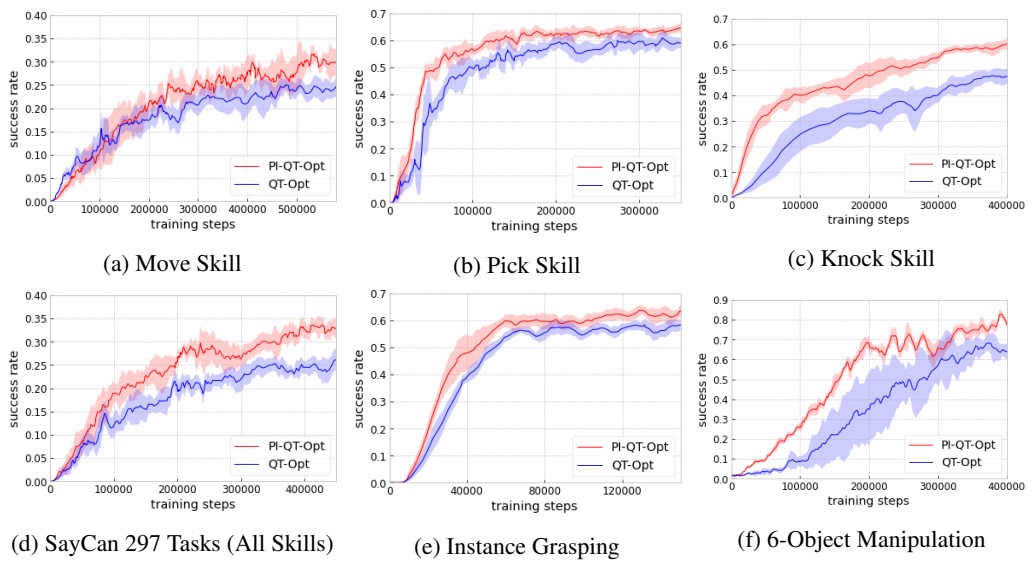

| (a) Move Skill | (b) Pick Skill | (c) Knock Skill |
| (d) SayCan 297 Tasks (All Skills) | (e) Instance Grasping | (f) 6-Object Manipulation |

Figure 3: Performance (success rate) on tasks that are used for agent training.

means that the current action is computed while the previous action is still executing. We provide additional implementation details in the Appendix B.

## 4.2  Reducing the Sim-to-Real Gap with CycleGAN

We train our PI-QT-Opt and QT-Opt models in simulated environments that roughly match the real world evaluation environments. In order to reduce the simulation-to-real (sim-to-real) gap when deploying policies on real robots, we train a RetinaGAN [57] model, a version of CycleGAN [58], to transform simulated robot images to look more realistic while preserving general object structure, following the sim-to-real setup in [14]. This enables our method to train purely on RetinaGAN-transformed simulation images and directly transfer to the real world. We apply this image transformation to all the SayCan models, which we evaluate in both simulation and the real world.

## 4.3  Training and Evaluation Protocols

For each of the experimental settings in Section 4.1, we create a sparse binary reward (success or failure) in simulation based on ground truth object poses. For each environment, we follow the large scale asynchronous distributed training procedure in [12] and train QT-Opt agents and PI-QT-Opt agents with the same hyperparameters with a batch size of 4096 using 16 TPUv2. In simulation, we evaluate models with 700 episodes and compute their success rates. For the real-world evaluations, we test each policy on 50 episodes of standardized starting scenarios for a fair comparison between policies. More details are described in Appendix B.6 and Appendix B.7.

## 5  Experimental Results

We discuss experimental results on the manipulation tasks introduced in Section 4.1 in Section 5.1 and Section 5.2. We report mean and one standard deviation of success rate over 3 training runs with different random seeds for each model. We analyze the relationship between predictive information and agent performance in Section 5.3 and information compression in Appendix E.

## 5.1  Evaluation in Simulated Environments

**Performance on training tasks.**   We learn PI-QT-Opt and QT-Opt models for each of the experiment settings introduced in Section 4.1. Figure 3 shows evaluation results on tasks that are used for agent training in simulation. We can see that PI-QT-Opt consistently outperforms QT-Opt in all settings throughout training, improving the move model by 20% and the 297-tasks model by 25%

relatively for example. This empirically validates our hypothesis that training with the predictive information auxiliary loss leads to better and more efficient learning of general-purpose agents.

**Zero-shot transfer to unseen tasks.** We use the held-out SayCan tasks to evaluate zero-shot transfer of PI-QT-Opt and QT-Opt models. A SayCan task is a composition of a skill and a set of target objects (1-2) that the robot should interact with, as described in Section 4.1. Here, we consider two types of zero-shot transfer: (1) the task is never seen during training, but the target objects have been seen during training in other tasks, and (2) not only the task but the object is never seen during training. For (1), we evaluate the move skill models on held-out move tasks, and the 297-tasks models on held-out tasks of all skills[3]. For (2), we evaluate the pick/knock skill models on held-out pick/knock tasks. As demonstrated in Figure 4, PI-QT-Opt outperforms QT-Opt in all the settings, showing that using the predictive information auxiliary loss leads to better zero-shot transfer to novel task compositions and unseen objects. The 297-tasks model, for instance, is improved by 28% relatively.

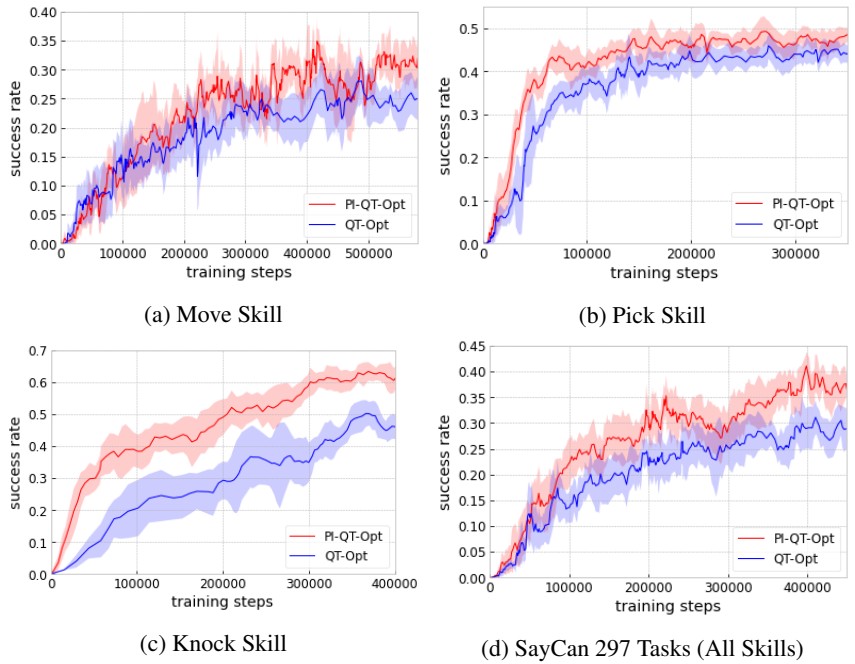

(a) Move Skill

(b) Pick Skill

(c) Knock Skill

(d) SayCan 297 Tasks (All Skills)

Figure 4: Performance (success rate) on unseen novel tasks.

## 5.2  Evaluation in the Real World

We directly deploy our PI-QT-Opt and QT-Opt models trained in simulation for SayCan skills on a real robot in the SayCan kitchen environment. As described in Section 4.1, these models use image-based task context conditioning that requires information about the initial object locations (Section 3.2). Unlike the simulated environments, the ground truth locations are not available in the real environments. Therefore, we use a VILD [59] model to detect objects conditioned on object names to locate the target objects associated with the names. When the VILD model fails to detect the target objects, we adjust the scene slightly until VILD succeeds and then continue with the evaluation. This allows us to evaluate the policy performance exclusively. We evaluate the SayCan Pick, Move, and Knock models; the results are presented in Table 1, showing a clear advantage for PI-QT-Opt. This empirically supports our hypothesis that using the predictive information auxiliary loss enables better performance when transferring to the real world.

---

[3]The Move Skill training task set contain 246 tasks and each involves 2 objectives. Due to the nature of this task set, it is difficult to avoid seen objects in held-out tasks. Therefore, we only use it for type (1) zero-shot transfer but not type (2).

Table 1: Evaluations on the real robot (mean and standard deviation over 3 evaluations)

| Task | PI-QT-Opt Success Rate | QT-Opt Success Rate | Relative Change |
|---|---|---|---|
| SayCan Move | **22.9 ± 8.4%** | 13.93 ± 3.2% | +64.4% |
| SayCan Pick | **42.0 ± 9.9%** | 28.7 ± 8.0% | +46.6% |
| SayCan Knock | **54.6 ± 2.4%** | 36.2 ± 11.6% | +50.7% |

### 5.3 How does Predictive Information Relate to Agent Performance?

A core hypothesis of this work is that the ability to model what will happen next is critical to success on control tasks. This ability can be quantified by the amount of predictive information, $I(X,Y)$, the agent's representation captures. In this section, we analyze the SayCan 300-task PI-QT-Opt model. We compare the estimates of $I(X,Y)$, $\mathbb{E}[\log b(z|y) - \log \frac{1}{K}\sum_{k=1}^{K} b(z|y_k)]$ (from Equation (2)) in successful and failed episodes versus TD-error in Figure 5[4]. We can observe that the amount of predictive information is generally higher in successful episodes, and that episodes with high TD-errors have much lower predictive information and are always failures.

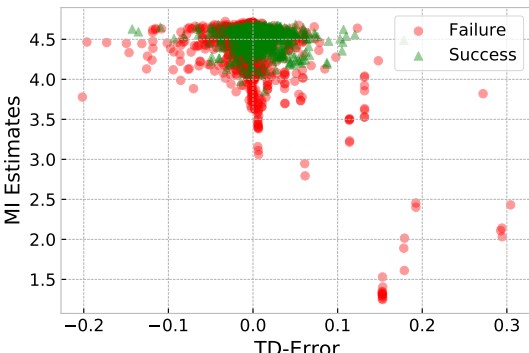

Figure 5: Predictive information estimate versus TD-error. Each point is averaged over an episode. The predictive information is the mutual information (MI) between the past and future.

## 6 Conclusion

We have shown that using the predictive information auxiliary with a QT-Opt agent, i.e. PI-QT-Opt, results in faster training, higher final performance, and better generalization to unseen tasks for a single generalist agent trained on hundreds of tasks. We have also shown that PI-QT-Opt and QT-Opt can be made to support multiple tasks by adding simple task conditioning. Our system, PI-QT-Opt, is a generalist agent capable of solving hundreds of real-world tasks in a simple kitchen environment, in spite of having only been trained in simulation.

**Limitations.**  We focused on a single robotic arm and gripper, so we cannot speculate on how well our approach would work on different robotic setups. We limited our real-world experiments to simple, carefully-controlled "kitchen" settings, and can say nothing about performance on different types of environments. Most importantly, our models do not have any safety guarantees, and we did not attempt to evaluate how they would perform in the presence of other agents, such as humans or animals. Using these models in settings where there are other agents could lead to injury or death. These limitations may be mitigated in future research by working with a range of robot platforms, expanding the breadth of tasks considered, including other agents in the environments during training and evaluation, and integrating safety systems.

---

[4]For these predictive information analyses, we collect data for each task with a converged policy, and use a batch size of 128, which corresponds to an $I(X,Y)$ upper bound of $\log 128 = 4.852$ in order to fit these analyses into one machine, while the distributed training batch size is 4096.

**Acknowledgments**

The authors would like to thank Alex Herzog, Mohi Khansari, Daniel Kappler, Peter Pastor for adapting infrastructure and algorithms for the image-based task context from generic to instance specific grasping, Sangeetha Ramesh for leading robot operations for data collection and evaluations for the VILD model training, and Kim Kleiven for leading the waste sorting service project that constitutes the framework for training and deployment of the instance grasping task set, including defining benchmark and protocol. We thank Jornell Quiambao, Grecia Salazar, Jodilyn Peralta, Justice Carbajal, Clayton Tan, Huong T Tran, Emily Perez, Brianna Zitkovich and Jaspiar Singh for helping administrate real-world robot experiments. We would also like to thank Sergio Guadarrama and Karol Hausman for valuable feedback.

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
