# OpenReview forum: "PI-QT-Opt: Predictive Information Improves Multi-Task Robotic Reinforcement Learning at Scale"
_robot-learning.org/CoRL/2022/Conference — CoRL 2022 Poster_

### Official Review · Reviewer_UcuR · 2022-07-27

**Originality:** Fair
**Technical Quality:** Good
**Clarity Of Presentation:** Very Good
**Impact:** 3

**Recommendation:**

Weak Reject: I recommend rejecting the paper, but will not argue for my recommendation if the majority of other reviewers have a different opinion.

**Summary:**

This paper proposes to add an auxiliary loss of predictive information to a model-free RL algorithm, QT-Opt. This additional loss helps shape the representations, which leads to efficient training and better transfer. The exhaustive simulated and real-world experiments demonstrate that QT-Opt combined with the auxiliary predictive information loss learns faster, generalizes better, and zero-shot transfers to the real world by training with sim-to-real translated images.

**Issues:**

* The proposed approach shows marginal improvements over the baseline algorithm. Also, it is not clear whether the proposed predictive information loss is critical for representations necessary for learning diverse manipulation tasks. It could be more convincing if the proposed PI-Qt-Opt is compared against other well-known representation learning approaches (e.g. pre-trained R3M networks [1], Transporter networks [2], ATC [3]).


[1] Nair et al. R3M: A Universal Visual Representation for Robot Manipulation, 2022

[2] Zeng et al. Transporter Networks: Rearranging the Visual World for Robotic Manipulation, CoRL 2020

[3] Stooke et al. Decoupling Representation Learning from Reinforcement Learning, ICML 2021

**Quality Of The Limitations Section:**

Additional details required

**Reviewer Expertise:**

2: The reviewer is willing to defend the evaluation, but it is quite likely that the reviewer did not understand central parts of the paper

**Robotics Focus:**

Sufficient demonstration on hardware

**Strengths And Weaknesses:**

### Strenghts

* Even though the proposed method is built upon existing methods (QT-Opt, multi-task learning, predictive auxiliary loss, and CycleGAN), the paper proposes a practical integration of these components to achieve a multi-task policy that transfers to the real world.

* The experiments in simulation and the real world demonstrate the effectiveness of using predictive information for representation learning over QT-Opt.


### Weaknesses

* The baseline performance of QT-Opt shows comparable results with the proposed method, PI-QT-Opt, except for Knock skills.

* The idea of supervising representations with predictive information itself is not new, which is one of the major contributions of the paper.

* The task definitions in the experiments are limited to simply specifying the target objects and the target regions. It is questionable whether this approach is scalable to more complex tasks.

### Questions and suggestions

* In Figure 2e, the robot seems to reach the orange, but the robot does not seem to move the orange near the purple area. What is the robot supposed to do? Similarly, Figure 2g is also unclear.

* One of the main claims of the paper, PI-QT-Opt solves up to "297 tasks", can be misleading. Before reading the details of the tasks, these tasks are likely to be interpreted as a lot of diverse types of manipulation tasks (e.g. pushing, inserting, pouring, folding, screwing) rather than pick-and-place with 200+ variations. This should be written more clearly.

**Summary Of Recommendation:**

The proposed method integrates many existing techniques to improve a multi-task robotic manipulation agent, which also zero-shot transfers to the real-world robot. However, the proposed method is incremental to the prior work PI-SAC and QT-Opt and the improvement is marginal compared to the baseline. Thus, I stand for weak rejection.


**After rebuttal**

I appreciate the authors' detailed answers. I appreciate that this paper demonstrates the effectiveness of "predictive information loss" for multi-task RL and sim-to-real transfer, and the experiments are impressive. Although this reassures the effectiveness of the predictive model loss for representation learning, the technical novelty seems incremental. Also, it could be better if the authors provided an updated paper addressing all the reviewers' comments, which may help understand the paper better. Thus, I am not confident to recommend accepting this paper.

---

> ### Author Response · Authors · 2022-08-25
> **Author Response to Reviewer UcuR**
>
> Thank you for your thoughtful feedback! Please let us know whether the following responses address your concerns and whether there are any points remaining, which we would be happy to discuss more.
>
> **Significance of the results**
>
> Looking at Figure 3 (on training tasks in simulation), we can see that:
> Move: 25% &rarr; 30% (20% relative improvement)
> Pick: 60% &rarr; 65% (8.3% relative improvement)
> Knock (updated): 48% &rarr; 60% (25% relative improvement)
> Note that we will update the knock results with longer training as included in the PDF attached to this response, as other reviewers asked.
>
> We can also look at the real evaluation results. For example, Table 2 (the SayCan-297 model) shows
> Move: 12.5% &rarr; 27.1% (117% relative improvement)
> Pick: 33.3% &rarr; 41.7% (20% relative improvement)
> Knock: 40% &rarr; 54.2% (26% relative improvement)
>
> We think the relative improvements are significant in all figures and tables, and will add relative improvements explicitly to the paper to emphasize the significance.
>
> **The supervising representations with predictive information itself is not new, which is one of the major contributions of the paper**
>
> We do not claim that we introduce the predictive information representation auxiliary, for which we cite [2]. For the major contributions of the paper, please refer to “**The contributions and value of this work**” in our response to the meta review, in addition to what we have in the introduction section.
>
> **How to extend the image-based task context to more complicated tasks**
>
> The tasks we selected are in a structure language form of **skill + target object sets**. How to extend to more complicated, free-form language commands is indeed not trivial. To support complicated commands, we could consider encoding the task description with a language model and provide object detection network activations instead of the final detection results, which can be considered a generalized version of our method. We will add these to the paper.
>
> **Figure 2**
>
> We will add a description for each frame sequence in Figure 2 and polish it more as suggested. We agree that will help readers to understand.
>
> **Tasks are likely to be interpreted as a lot of diverse types of manipulation tasks (e.g. pushing, inserting, pouring, folding, screwing) rather than pick-and-place with 200+ variations. This should be written more clearly.**
>
> We use the word **skill** for pushing, inserting, pouring, folding, screwing, etc. In Section 3.2 (line 148, right before Section 4.1 where we introduce manipulation tasks), Section A.2, and Section A.6, we explained that, in our setting, a manipulation task involves a robot skill (e.g. move, pick, knock) and a set of objects of interest. We will add a note in the introduction where “297 tasks” appears for the first time in the main text.
>
> **Compare to other representation learning approaches**
>
> Let’s first discuss the three papers referred to in the review. R3M’s main contribution is empirically demonstrating that representations pre-trained on diverse human video datasets like Ego4D can enable efficient downstream policy learning for robotic manipulation, which is orthogonal to and out of the scope of this work. Transporter Network relies on a depth camera, which is not what we are interested in. ATC [5] (ICML 21’) is essentially the same representation learning approach as PI-SAC [2] (NeurIPS 20’) but without discussing information compression. Both papers explored using predictive information as an auxiliary and decoupled representation learning objectives. As we base our work from PI-SAC, which uses a more general objective and was proposed prior to ATC, we think it is not necessary to discuss ATC.
>
> Broadly speaking, comparisons to representation approaches (e.g. PI-SAC vs CURL, discriminative vs generative) have been extensively explored and discussed in prior work such as [2] and [15]. Adding such experiments here would add substantial complexity into this work which already does very large-scale experiments. Therefore, we think it is preferable to focus on very large-scale multi-task learning and sim-to-real in the context of robot manipulation.

---

### Official Review · Reviewer_G4sr · 2022-07-29

**Originality:** Fair
**Technical Quality:** Good
**Clarity Of Presentation:** Very Good
**Impact:** 4

**Recommendation:**

Weak Accept: I recommend accepting the paper, but will not argue for my recommendation if the majority of other reviewers have a different opinion.

**Summary:**

This paper improves the QT-Opt algorithm by adding a predictive information axillary loss. The proposed algorithm PI-QT-Opt achieves better performance (task success rate) over various tasks in both simulator and real-world robotic testbed. The extensive experiments validate the advantage of the predictive information axillary loss.

**Issues:**

- Please clarify the difference between the proposed PI-QT-Opt algorithm and [2].
- Please provide more explanation of the tasks in line 232  "(2) not only the task but the object is never seen during training.". Please define "the task ... is never seen during training".
- Please point out what could limit PI-QT-Opt to achieve better results in table 1 in the limitation section. Though PI-QT-Opt is better than QT-Opt, there is room for improvement.
- How do you think what are the advantages/ disadvantages of the predictive information, compared with model-based RL algorithms? Model-based RL algorithms also utilize the information between the current state action tuple and the future state reward tuple.

**Quality Of The Limitations Section:**

Additional details required

**Reviewer Expertise:**

3: The reviewer is fairly confident that the evaluation is correct

**Robotics Focus:**

Sufficient demonstration on hardware

**Strengths And Weaknesses:**

Strengths
- The extensive experiments validate the advantage of the predictive information axillary loss.
- The predictive information axillary loss shows advantages in all the tasks in the figure. 3, 4. This indicates the benefit of predictive information exits in vast robotic tasks.

Weaknesses
- The proposed algorithm PI-QT-Opt is very similar to [2]. If it is true, this paper will be more about an empirical study of applying [2] to robotic manipulation. (Please correct me if I am wrong)

**Summary Of Recommendation:**

The paper proposed an algorithm PI-QT-Opt that incorporates a predictive information axillary loss. The benefit of predictive information axillary loss is been validated by extensive experiments in the paper. Whoever, the proposed algorithm PI-QT-Opt is very similar to [2]. The similarity reduces the novelty of the paper.

---

> ### Author Response · Authors · 2022-08-25
> **Author Response to Reviewer G4sr**
>
> Thank you for your thoughtful review! Please let us know whether the following responses sufficiently address your concerns and whether there are any points remaining, which we would be happy to address.
>
> **Difference to PI-SAC [2]**
>
> For differences in contributions, please refer to “**The contributions and value of this work**” in our response to the meta review. We will also refer to the difference to PI-SAC more explicitly in describing the design choices of PI-QT-Opt in the paper.
>
> **Line 232, definition of unseen tasks**
>
> As we explained in Section 3.2, Section A.2, and Section A.6, in our definition, **a task = a skill (pick, move, etc) + a set of objects**, such as “pick obj1” and “move obj1 to obj2”.
>
> *“(1) the task is never seen during training, but the target objects have been seen during training in other tasks”* means that, for example, the model could have seen the tasks “move obj1 to obj2” and “move obj1 to obj3” during training but not “move obj2 to obj3” which we use as a test task.
>
> *“(2) not only the task but the object is never seen during training”* means that, for example, the model could have seen “pick obj1” and “pick obj2” during training but not “pick obj3” which we use as a test task (in other words, obj3 is never seen during training).
>
> **Mention that there is room for improvement for PI-QT-Opt in the limitation section**
>
> We will add it in the limitation section as suggested.
>
> **The advantages and disadvantages of the predictive information, compared to Model-Based RL algorithms**
>
> There are two components in many model-based RL algorithms: the generative observation and reward prediction and latent rollout.
>
> The contrastive InfoNCE for predictive information estimation approximates the generative prediction distribution p(future|past)$ [2, 47]. Doing maximum likelihood for p(future|past), e.g. with a deconvolutional network, is equivalent to maximizing the predictive information [2, 47]. In Section G of [2], there are extensive experimental results comparing InfoNCE and explicit future state-reward prediction with a decoder network. Compared to the contrastive approach, deconvolution is often more expensive to train and converges more slowly as shown in [2]. For PI-QT-Opt, we make the forward encoder and online Q-network share parameters, and the backward encoder and the target Q-network share parameters, so we only need very little additional parameters for the MLP heads, compared to having a decoder network. The downside of the contrastive approach is that you need sufficiently large batch size to supply meaningful negative samples to estimate the partition function, although that has not been a problem in our case. Note that it is possible for both contrastive and generative approaches to learn compressed representations, as demonstrated in [2]. [Dreamer](https://arxiv.org/abs/1912.01603), which is a model-based approach with both generative prediction and latent rollout, also learns a compressed representation by training the internal VAE while targeting a particular information budget.
>
> Latent rollout (modeling p(future_latent|current_latent, action)) is orthogonal to learning a predictive information representation. It is possible to learn latent representations by capturing the predictive information with InfoNCE and perform latent rollout. [Okada and Taniguchi](https://arxiv.org/abs/2007.14535) showed that this can empirically work well. However, at the moment, the only approaches that we are aware of that can work in the multitask setting on our real robot setting are based on QT-Opt, so we have not explored using a latent rollout approach yet.

---

### Official Review · Reviewer_jMwU · 2022-08-01

**Originality:** Good
**Technical Quality:** Very Good
**Clarity Of Presentation:** Excellent
**Impact:** 4

**Recommendation:**

Weak Accept: I recommend accepting the paper, but will not argue for my recommendation if the majority of other reviewers have a different opinion.

**Summary:**

This work studies the effect of improving RL training performance with an auxiliary loss for the task of finding the mutual information between past and future states. The specific technique for the auxiliary task is using a Conditional Entropy Bottleneck, which is a representation learning task previously shown to improve RL training in "predictive information accelerates learning in RL" [2]. Unlike in [2], this paper's focus is on augmenting QT-Opt with the auxiliary loss (as opposed to augmenting SAC) for the purpose of vision-based task-conditioned robotic object manipulation (as opposed to simpler simulated continuous tasks), with task conditioning allowing a single agent to be trained for hundreds of different tasks. The main contribution of this work are the empirical results of applying the ideas from [2] to this different setting, as well as the demonstration of multi-task learning by task condition through images or language.

**Issues:**

As per my summary and noted weaknesses, I think presenting further analysis with additional insights would strengthen this work. Beyond that, more discussion of task-conditioning and some iteration on Figure 2 would be benficial.

**Quality Of The Limitations Section:**

Limitations are addressed clearly

**Reviewer Expertise:**

5: The reviewer is absolutely certain that the evaluation is correct and very familiar with the relevant literature

**Robotics Focus:**

Sufficient demonstration on hardware

**Strengths And Weaknesses:**

Strengths:
* The paper is well written and is easy to follow. For the most part there is no missing information in the explanation of related work, technical details, and experimental setup details.
* The dual contributions of extending the findings of [2] to complex task-conditioned manipulation and of demonstrating a new technique for task-conditioning QT-Opt are of significant value to the field.
* The set of experiments and the results observed convincingly demonstrate this auxiliary loss does help with training, and more significantly appears to improve real-world performance by a large margin. While this primarily shows
* The papers notes some significant limitations and how they might be addressed in future work

Weaknesses:
* Results for the knock skills are confusingly different from the other ones, with lesser granularity and significantly larger benefits from using the auxiliary loss.
* Apart from section 5.3, there is not much analysis of the results. Section D from the appendix does present some ablation results, which I feel would have been better in the main paper.
* The (e), (f), and (g) rows of Figure 2 have really small images that are hard to understand.
* It is stated that "With our best effort, we were unable to get SAC and PI-SAC working on our manipulation tasks." Given this work specifically builds on [2], I would have liked to see more discussion or analysis of why QT-Opt with PI worked better than PI-SAC.
* There is very little discussion of task-conditioning, and no citations of prior works on the topic. Given the second claimed contribution is "We show that simple forms of task conditioning are sufficient to allow QT-Opt", it feels like more discussion of related ideas and evaluation of how well task conditioning actually works are needed.

**Summary Of Recommendation:**

I believe this work presents valuable empirical results that involved a substantial amount of experimentation that shows the the key idea from  [2] applies to the robotics domain. However, given the contributions are primarily about empirical results rather than technical ideas, I would have liked to see significantly more analysis of the results that could provide further insights similar to the one in section 5.3.

---

> ### Author Response · Authors · 2022-08-25
> **Author Response to Reviewer jMwU**
>
> Thank you for your thoughtful feedback! Please let us know whether our following response fully addresses your concerns.
>
> **Knock experiments**
>
> In the PDF attached to this response, we include longer PI-QT-Opt and QT-Opt training curves up to 400k steps. The training and testing success rates are improved, but the absolute gap between PI-QT-Opt and QT-Opt remains similar and significant. We also performed real-world evaluation with the PI-QT-Opt and QT-Opt knock skill models trained for 400k steps. PI-QT-Opt achieves 57.1% success rate and QT-Opt achieves 48.6% success rate on the real robot. We will update the paper with these results.
>
> **Ablation on compression**
>
> We will attempt to fit the compression ablation in Section D into the main paper. If that turns out to still be difficult, we will briefly describe the findings and refer the reader to the appendix.
>
> **Figure 2(e)(f)(g)**
>
> We will add a description for each frame sequence in Figure 2 and polish it more as suggested. We agree that will help readers to understand.
>
> **QT-Opt vs SAC**
>
> We agree that having more discussion about QT-Opt vs SAC could be useful. Compared to SAC, the main advantage of QT-Opt is that the action selection is pure sampling-based (CEM), and thus does not require a gradient-learned actor as in SAC. This makes it possible to have complex and even dynamic action space and bounds without worrying about how to back-propagate gradients.
>
> This makes adding safety constraints simple. Every time when we sample an action, we can clip the action according to the action bound, which can change based on the safety constraints at each specific robot state. For a gradient-based actor, such clipping zero-outs gradients, making optimization challenging. We did try training SAC and it did not learn with safety-constraint action clipping. How to make a gradient-based actor work in our setting is still an open-ended research question, for which we don’t have a good answer at the moment. We will include these descriptions in the paper.
>
> **Task conditioning**
>
> For task conditioning, we discuss implementation details and design choices in Section A.2 in the appendix (due to space constraint of the main paper) and cited [“Multi-Task Domain Adaptation for Deep Learning of Instance Grasping from Simulation”](https://arxiv.org/abs/1710.06422) as a prior work (Appendix [4]). In contrast to the prior work which uses pixel-accurate object segmentation masks, we use a square image mask, agnostic of the object size, to indicate the center of the object. As we explained in Section A.2, in the real world, predicting pixel-accurate masks can be sensitive to many conditions including lighting, occlusions, the angle of view, and other perturbations. In our early experiments, we did not find performance differences in simulation between the two approaches, but on real hardware, for our object set, the segmentation mask accuracy is significantly worse and leads to a large sim-to-real gap. As for how well task conditioning works, our claim is simply that it is sufficient to solve very large-scale multi-task robot manipulation, whereas the data routing approach (MT-Opt [10]) is difficult to scale to so many tasks (line 85-89).

---

> > ### Author Response · Authors · 2022-08-26
> > **Response Update**
> >
> > We have added the real-world evaluation results of the 400k-step PI-QT-Opt and QT-Opt knock models in "Knock experiments" in the above response.

---

> > > ### Comment · Reviewer_jMwU · 2022-08-27
> > > **Final Response**
> > >
> > > Thank you for your response concerning my feedback and for updating the paper to reflect my suggestions - I feel that the weaknesses brought up in my review have been well addressed!

---

### Official Review · Reviewer_QJ7c · 2022-08-04

**Originality:** Good
**Technical Quality:** Very Good
**Clarity Of Presentation:** Excellent
**Impact:** 3

**Recommendation:**

Weak Accept: I recommend accepting the paper, but will not argue for my recommendation if the majority of other reviewers have a different opinion.

**Summary:**

The paper aims to train vision-based policies that can master a wide range of skills by learning from a large amount of diverse experience. The paper claims that learning a latent representation of the environment dynamics by modeling the predictive information (introduced by a prior work) result in policies that generalize better (zero-shot) to unseen tasks and for sim to real settings. The experimental results with up to 297 vision-based manipulation tasks (kitchen environment) support their claim.


**Issues:**

Figure 1: using s to denote both proprioceptive state and image observation can be misleading.

**Quality Of The Limitations Section:**

Additional details required

**Reviewer Expertise:**

4: The reviewer is confident but not absolutely certain that the evaluation is correct

**Robotics Focus:**

Sufficient demonstration on hardware

**Strengths And Weaknesses:**

Strengths:
* The experimental results seem to support that modeling the predictive information can improve generalization.
* It is a very well written and easy to read paper

Weaknesses:
* The theoretical contributions of the paper are minor.
* The improvement compared to QT-Opt is not significant. Here is a summary for figure 3:
Move skill: QT-Opt: 25%, PI-QT-Opt: 30%
Pick skill:  QT-Opt: 43%, PI-QT-Opt: 48%
Knock skill: QT-Opt: 19%, PI-QT-Opt: 38% - It is not convincing why QT-Opt performs so poorly compared to Pick skill since the Knock skill must be much easier than pick.
* It is not really clear how the outcome of this research can help other researchers to build more general policies in multi-task settings. It is difficult for other researchers to reproduce the setup or to have access to the computation power used in this experiment.

**Summary Of Recommendation:**

It is still hard to believe that modeling a latent representation using predictive information can help since the success rate is still very low for both PI-QT-Opt and QT-Opt.

---

> ### Author Response · Authors · 2022-08-25
> **Author Response to Reviewer QJ7c**
>
> Thank you for your thoughtful feedback! Please let us know whether the following responses fully address your concerns or whether there are any points remaining, which we would be happy to address.
>
> **Theoretical contributions are minor**
>
> Please refer to “**The contributions and value of this work**” in our response to the meta review.
>
> **The knock skill models**
>
> In the PDF attached to this response, we include longer PI-QT-Opt and QT-Opt training curves up to 400k steps. The absolute gap between PI-QT-Opt and QT-Opt remains similar and significant. The training and testing success rates are indeed improved and eventually become similar to or better than the pick model success rate. However, we note that the pick and knock success rates are not directly comparable, as they operate on different sets of objects. The knock skill target objects only include cans and bottles since other objects cannot be “knocked down” from an upright pose (see Section B for our object list), and there are 7 objects for training and 1 held-out object for testing. The pick tasks use all 17 objects, where 12 objects are used for training and 5 objects are held-out. We will include these descriptions and a list of tasks (skill + objects) for each skill in the paper to clarify.
>
> We also performed real-world evaluation with the PI-QT-Opt and QT-Opt knock skill models trained for 400k steps. PI-QT-Opt achieves 57.1% success rate and QT-Opt achieves 48.6% success rate on the real robot. We will update the paper with these results.
>
> **The significance of the results**
>
> Our summary of Figure 3 (on training tasks in simulation) is:
> Move: 25% &rarr; 30% (20% relative improvement)
> Pick: 60% &rarr; 65% (8.3% relative improvement)
> Knock (update results with 400k training steps): 48% &rarr; 60% (25% relative improvement)
>
> We can also look at the real evaluation results. For example, Table 2 (the SayCan-297 model) shows
> Move: 12.5% &rarr; 27.1% (117% relative improvement)
> Pick: 33.3% &rarr; 41.7% (20% relative improvement)
> Knock: 40% &rarr; 54.2% (26% relative improvement)
>
> We think the relative improvements are significant in all figures and tables, and will add relative improvements explicitly to the paper to emphasize the significance.
>
> **Access to computation power and reproducibility**
>
> We agree that it is difficult for researchers to reproduce our exact setup, since the robots the experiments ran on are still proprietary, and since the simulation environments we train on have not been released (in this case, that shortcoming is outside of our control). However, we think it’s not uncommon for robot learning research to involve custom robots and complex system building. Among such peer-reviewed publications include Lu et al. [12] (CoRL 2021), Jang et al. [24] (ICLR 2021), Ahn et al. [14] (RSS 2021 Workshop), etc. We do believe that our findings can generalize to similar multi-task robot manipulation settings, and we include clear descriptions of our robot and tasks to make sure the readers understand our experimental setup.
>
> We also agree that we spent a great deal of time and computation on this research, and even with access to the same robotic platform and the same simulation environments, many research labs would have difficulty reproducing our full suite of experiments for now. However, as we explained in the introduction and the response to the meta review, one of our main contributions is to see whether the gains of the predictive information exist for multitask Q-learning, sim-to-real transfer, and unseen tasks at a large scale. This contribution itself is inherently difficult to reproduce for smaller labs.  On the other hand, historically, the computation power available to everyone has increased dramatically over time, so we expect the computation we have may be available to many more researchers in the relatively near future.
>
> **Notations in Figure 1**
>
> We will add subscripts to differentiate proprioceptive state and image observation in Figure 1.

---

> > ### Author Response · Authors · 2022-08-26
> > **Response Update**
> >
> > We have added the real-world evaluation results of the 400k-step PI-QT-Opt and QT-Opt knock models in "The knock skill models" in the above response.

---

### Meta-Review · Area_Chair_5CDz · 2022-08-15

**Recommendation:** Accept (Poster)
**Confidence:** 4

**Metareview:**

Strengths:
- All the reviewers have found that the experimental results of the paper is convincing.
- All the reviewers have found the technical approaches in the paper making sense.
- The paper is well-written and well-structured.


Weaknesses:
- Most reviewers have pointed out limited theoretical contributions as this article basically deals with practical applications of an existing methodology in [2]. => Emprical contributions have been strenghthened during the rebuttal phase.

**Best Paper Nomination:**

No

---

> ### Author Response · Authors · 2022-08-25
> **The contributions and value of this work**
>
> **The contributions and value of this work**
>
> Our contributions are empirically focused, as we explained in the introduction (line 47-57). However, whether the predictive information auxiliary can improve multi-task learning at a very large scale and whether the improvement can transfer to the real-world domain are not trivial questions in our opinion. Possible failure patterns that we expected before starting the project include:
> - As the number of tasks increase to such a large scale, the Q-network is required to be more general, and the regularization benefits of the predictive information auxiliary for preventing overfitting [R1] could disappear.
> - The predictive information captured in simulated experience is not meaningful enough to transfer the improvement to real-world robots.
> - The predictive information may not help the policy to generalize to unseen objects.
>
> All of these have not been answered in prior work, which focus on single-task, simulated environments. Fortunately, all of these concerns proved to be false, and the predictive information gave substantial boosts to the performance of the system. This work confirms the theoretical benefits of the predictive information to generalization specifically shown in “Predictive Information Accelerates Learning in RL” [2] in a much more challenging multitask RL setting, as well as for zero-shot transfer to the real world and unseen tasks. We think these are indeed useful new results for the machine learning and robot learning communities.
>
> Additionally, we made contributions in designing task contexts, especially the image-based one, to make it possible to tackle multitask RL for robot manipulation, generalization to new tasks, and sim-to-real transfer at a very large scale.
>
> We will include the discussion above in the paper, and hope that helps clarify our contributions and values.
>
> [R1] The Value-Improvement Path: Towards Better Representations for Reinforcement Learning. https://arxiv.org/abs/2006.02243.